# Soil Nitrogen in Response to Interseeded Cover Crops in Maize–Soybean Production Systems

**Yesuf Assen Mohammed [1],\*, Swetabh Patel [2], Heather L. Matthees [3], Andrew W. Lenssen [2]**, **Burton L. Johnson [4], M. Scott Wells [5], Frank Forcella [1], Marisol T. Berti [4] and Russ W. Gesch [1],\***

1   North Central Soil Conservation Research Laboratory, Agricultural Research Service, United State Department of Agriculture, 803 Iowa Ave, Morris, MN 56267, USA; forcellafrank@gmail.com

2   Department of Agronomy, Iowa State University, 2104 Agronomy Hall, Ames, IA 50011, USA; swetabh@iastate.edu (S.P.); alenssen@iastate.edu (A.W.L.)

3   WinField United, Land O'Lakes, 4001 Lexington Ave, Arden Hills, MN 55112, USA; heather.matthees@gmail.com

4   Department of Plant Sciences, North Dakota State University, Fargo, ND 58102, USA; burton.johnson@ndsu.edu (B.L.J.); marisol.berti@ndsu.edu (M.T.B.)

5   Department of Agronomy and Plant Genetics, University of Minnesota, 1991 Upper Buford Circle, 411 Borlaug Hall, St. Paul, MN 55108, USA; mswells@umn.edu

\*   Correspondence: yesuf.mohammed@usda.gov (Y.A.M.); russ.gesch@usda.gov (R.W.G.)

**Abstract:** Improved agronomic management strategies are needed to minimize the impact that current maize (*Zea mays* L.) and soybean (*Glycine max* (L.) Merr.) production practices have on soil erosion and nutrient losses, especially nitrogen (N). Interseeded cover crops in standing maize and soybean scavenge excess soil N and thus reduce potential N leaching and runoff. The objectives were to determine the impact that pennycress (*Thlaspi arvense* L.) (PC), winter camelina (*Camelina sativa* (L.) Crantz) (WC), and winter rye (*Secale cereale* L.) (WR) cover crops have on soil N, and carbon (C) and N accumulation in cover-crop biomass. The cover crops were interseeded in maize at the R5 growth stage and in soybean at R7 in four replicates over two growing seasons at four locations. Soil and aboveground biomass samples were taken in autumn and spring. Data from the maize and soybean systems were analyzed separately. The results showed that cover crops had no effect on soil $NH_4^+$-N under both systems. However, winter rye decreased soil $NO_3^-$-N up to 76% compared with no-cover-crop treatment in the soybean system. Pennycress and WC scavenged less soil N than WR. Similarly, N and C accumulation in PC and WC biomass were less than in WR, in part because of their poor growth performance under the interseeding practice. Until PC and WC varieties with improved suitability for interseeding are developed, other agronomic practices may need to be explored for improving N scavenging in maize and soybean cropping systems to reduce nutrient leaching and enhance crop diversification.

**Keywords:** cover crops; camelina; pennycress; soil nitrate

## 1. Introduction

Maize and soybean are the two most economically important crops in the upper Midwest US. Despite short-term economic advantages of growing these crops, recurrent practices used for their production in the upper Midwest US have led to unintended environmental consequences including water pollution resulting from off-site nutrient losses, mainly nitrogen [1–3]. Erosion and runoff from cropland are major sources of nutrient loading to lakes and reservoirs [4,5]. Maize and soybean production contribute up to 52% of the nitrogen loading into the Gulf of Mexico creating hypoxia

and endangering marine life [6–8]. Water quality degradation due to nitrate contamination is critical. Although the US Environmental Protection Agency (EPA) established the Safe Drinking Water Act (SDWA) in 1975, which set a standard limit of 10 mg $L^{-1}$ for nitrate-N [9], nitrate contamination of surface and ground water from agricultural land is still a common problem [6]. For instance in Iowa, from 2003 to 2016, maize grain harvested area, maize grain yield (kg $ha^{-1}$), and nitrate loads to streams increased by 16, 47, and 29%, respectively [10,11]. Increased N fertilizer application rates for maize production but with reduced N use efficiency in the last few decades [12] may explain the increased nitrate loads to streams. This is aggravated in the absence of growing crops immediately following maize harvesting which would enhance residual nutrient uptake and minimize runoff. Increased stream $NO_3^-$-N loads have been indicated as a strong driver for the Gulf of Mexico hypoxic area increase [10]. Growing crops following maize harvesting to cover the soil and enhance residual nutrient uptake has been suggested as a solution to reduce nutrient loads to streams [10]. Cover crops can scavenge soluble N to a soil depth of 1.8 m [13]. Adoption of cover crops that could diversify the cropping systems could bring additional ecosystem services [13,14]. Furthermore, the lack of crop biodiversity due to extensive use of the maize–soybean rotation, has been linked to the decline in pollinators and other beneficial insects [2,14]. Limited crop diversification costs soybean producers in four U.S. states (Iowa, Michigan, Minnesota, and Wisconsin) an estimated USD 58 million $y^{-1}$ in reduced yield and increased pesticide use [2].

The use of different cover-crop species other than maize and soybean could diversify the maize–soybean-dominated cropping systems in the upper Midwest US. This could decrease nutrient losses from runoff and leaching and could provide feed for pollinators [13,15,16]. Despite understanding the various benefits of growing cover crops and increased promotional efforts, cover-crop adoption has been slow [17–19]. Lack of attractive short-term economic returns to growers and limited time for cover-crop establishment following maize and soybean harvest are major reasons for slow adoption [17,18,20]. Therefore, alternative cover crops and seeding practices are needed to provide immediate economic incentives to growers and extend the time necessary to establish cover crops in maize–soybean production systems.

Winter rye is arguably the most popular cover crop in the US [21,22]. A recent study indicated that current cost-share payments by the government will not incentivize widespread rye adoption as a cover crop [17]. Moreover, maize following WR can suffer due to diseases and excessive water use by rye [23–25]. These problems show the importance of seeking alternative cover crops to WR. Winter camelina and PC, both winter annual oilseeds, have recently received considerable attention as alternative cover crops in maize–soybean cropping systems to provide soil cover, add income, increase plant diversity, and provide additional ecosystem services such as pollen and nectars for insects [26–28].

Successful establishment of PC and WC as covers following short-season small grains, and the ecosystem services they offer is well documented [29]. However, establishment of these oilseeds following maize or soybean grain harvest in the upper Midwest latitude is problematic because of the short period between harvest and winter freezing [30–32]. Therefore, alternative practices to establish these crops in maize–soybean systems are needed. Previous research showed that WC interseeded on the same sowing date as maize decreased maize yield due to competition [30]. To minimize this competition, PC and WC could potentially be interseeded during the late reproductive stages of maize and soybean. Our previous results showed that WC and PC could be interseeded late in the reproductive stages of maize and soybean without affecting their yields [33]. Similar studies showed that maize yield was unaffected when similar cover crops were interseeded at the V2 to V7 growth stage of maize [18,20]. However, the impacts of interseeded PC and WC on soil N, cover-crop biomass C, and N accumulation are limited. A better understanding of the ability of PC and WC to scavenge soil N when interseeded in standing maize and soybean as cover crops is needed. Therefore, the objectives of this study were to determine the effects of interseeded PC, WC, and WR on soil $NH_4^+$-N and $NO_3^-$-N content, and to evaluate the difference in cover-crop N and C biomass accumulation.

## 2. Materials and Methods

### 2.1. Study Location

Field experiments were conducted at four locations in 2016/2017 and 2017/2018. The four locations were Ames (42°00′ N, −93°44′ W, and 326 m a.s.l ) in Iowa; Morris (45°40′ N, −95°48′ W, and 344 m a.s.l) and Rosemount (44°42′ N, −93°03′ W, and 284 m a.s.l) in Minnesota; and Prosper (46°58′ N, −97°03′ W, and 284 m a.s.l) in North Dakota, USA. The soils at each site were: Clarion loam (fine-loamy, mixed, superactive, mesic Typic Hapludolls) and Webster clay loam (fine-loamy, mixed, superactive, mesic Typic Endoaquolls) in Ames; Hokans (fine-loamy, mixed, superactive, frigid Calcic Hapludolls)-Svea (fine-loamy, mixed, superactive, frigid Pachic Hapludolls) complex in Morris; Waukegon silt loam (fine-silty over sandy or sandy-skeletal, mixed, superactive, mesic Typic Hapludolls) in Rosemount; and Kindred silt loam (fine-silty, mixed, superactive, frigid Typic Endoaquolls) and Bearden silt loam (fine-silty, mixed, superactive, frigid Aeric Calciuaquolls) in Prosper. Monthly total precipitation and monthly mean air temperature for each location are presented in Table 1. Weather data were obtained from nearby weather stations at each location. The three-year average (2016 to 2018) weather data showed that Ames was generally wetter and warmer than other locations while Prosper was drier and cooler (Table 1).

**Table 1.** Monthly total precipitation (mm) and monthly mean air temperature (°C) from January to December from 2016 to 2018 and long-term average (LTA) for the last 30 years at Ames (Iowa), Morris and Rosemount (Minnesota), and Prosper (North Dakota). Precipitation data for Prosper from November to March are missing. Weather data collected from nearby weather stations.

| Month | Ames | | | | Morris | | | | Prosper | | | | Rosemount | | | |
|---|---|---|---|---|---|---|---|---|---|---|---|---|---|---|---|---|
| | 2016 | 2017 | 2018 | LTA | 2016 | 2017 | 2018 | LTA | 2016 | 2017 | 2018 | LTA | 2016 | 2017 | 2018 | LTA |
| **Precipitation (mm)** | | | | | | | | | | | | | | | | |
| January | 15 | 47 | 33 | 17 | 7 | 13 | 4 | 19 | NA | NA | NA | - | 20 | 52 | 25 | 26 |
| February | 17 | 30 | 29 | 22 | 22 | 11 | 21 | 18 | NA | NA | NA | - | 14 | 16 | 28 | 23 |
| March | 38 | 79 | 63 | 50 | 21 | 12 | 20 | 35 | NA | NA | NA | - | 54 | 16 | 23 | 58 |
| April | 104 | 78 | 32 | 100 | 52 | 65 | 24 | 61 | 17 | 4 | 26 | 56 | 116 | 50 | 74 |
| May | 109 | 156 | 101 | 124 | 43 | 92 | 26 | 77 | 82 | 17 | 54 | 72 | 69 | 182 | 109 | 103 |
| June | 25 | 44 | 282 | 122 | 54 | 101 | 138 | 108 | 38 | 88 | 79 | 98 | 81 | 91 | 154 | 120 |
| July | 149 | 25 | 107 | 117 | 184 | 23 | 143 | 96 | 88 | 50 | 65 | 76 | 121 | 139 | 111 | 114 |
| August | 209 | 85 | 214 | 122 | 94 | 175 | 97 | 83 | 26 | 53 | 79 | 54 | 178 | 129 | 99 | 120 |
| September | 201 | 46 | 171 | 83 | 43 | 105 | 50 | 71 | 60 | 152 | 71 | 69 | 133 | 42 | 157 | 92 |
| October | 15 | 154 | 123 | 61 | 87 | 69 | 76 | 60 | 49 | 7 | 67 | 50 | 62 | 99 | 91 | 73 |
| November | 44 | 7 | 41 | 46 | 42 | 12 | 23 | 24 | NA | NA | NA | - | 45 | 2 | 38 | 53 |
| December | 30 | 4 | 67 | 29 | 33 | 7 | 28 | 18 | NA | NA | NA | - | 24 | 8 | 47 | 31 |
| **Air temperature (°C)** | | | | | | | | | | | | | | | | |
| January | −6.6 | −4.0 | −6.8 | −6.3 | −9.7 | −9.1 | −10.6 | −12.0 | −11.1 | −11.3 | −13.0 | −13.5 | −10.0 | −7.6 | −11.1 | −10.7 |
| February | −1.6 | 2.6 | −5.6 | −4.0 | −5.2 | −3.1 | −12.4 | −9.8 | −5.6 | −5.3 | −15.3 | −10.7 | −5.3 | −1.9 | −11.9 | −7.7 |
| March | 7.3 | 3.7 | 2.2 | 3.3 | 3.1 | −0.9 | −1.8 | −2.6 | 2.8 | −2.5 | −5.2 | −3.1 | 3.8 | −1.0 | −1.4 | −0.5 |
| April | 11.3 | 11.4 | 5.4 | 10.5 | 6.9 | 7.5 | 1.6 | 6.4 | 5.6 | 6.6 | 0.0 | 6.0 | 8.4 | 9.0 | 1.0 | 7.9 |
| May | 16.6 | 16.3 | 20.6 | 16.7 | 15.3 | 13.4 | 17.8 | 13.9 | 15.0 | 13.2 | 16.9 | 13.4 | 15.6 | 13.5 | 18.6 | 14.3 |
| June | 24.2 | 22.8 | 23.5 | 22.0 | 20.3 | 20.0 | 21.0 | 19.2 | 19.5 | 19.1 | 20.5 | 18.9 | 21.0 | 20.4 | 21.4 | 19.6 |
| July | 23.9 | 24.4 | 23.5 | 23.7 | 21.5 | 22.1 | 21.7 | 21.4 | 21.1 | 21.2 | 20.3 | 20.7 | 24.8 | 22.3 | 22.0 | 21.9 |
| August | 23.1 | 20.8 | 22.8 | 22.6 | 20.7 | 18.4 | 20.3 | 20.0 | 20.6 | 18.1 | 19.4 | 19.7 | 21.3 | 18.9 | 21.1 | 20.8 |
| September | 21.0 | 20.4 | 19.8 | 18.7 | 16.4 | 16.6 | 15.9 | 15.2 | 16.1 | 15.3 | 14.1 | 15.0 | 17.9 | 17.9 | 17.4 | 16.0 |
| October | 14.9 | 12.7 | 10.2 | 11.9 | 9.5 | 8.5 | 5.0 | 7.6 | 8.3 | 7.5 | 3.8 | 7.3 | 11.0 | 9.6 | 6.2 | 8.9 |
| November | 8.2 | 3.6 | −0.2 | 3.5 | 4.3 | −1.4 | −4.8 | −1.4 | 4.4 | −3.3 | −6.1 | −1.9 | 6.0 | −0.6 | −3.3 | 0.1 |
| December | −4.2 | −3.8 | −1.4 | −4.1 | −8.9 | −9.0 | −6.0 | −9.1 | −10.0 | −11.2 | −8.0 | −9.9 | −7.0 | −8.2 | −5.0 | −8.2 |

NA = Data not available.

### 2.2. Experimental and Treatment Designs

The experiment was conducted in two sets with three-year cropping sequences (maize–soybean–maize, and soybean–soybean–maize). However, for this report, only the effects of the first two-year sequences (maize–soybean sequence hereafter maize system and soybean–soybean sequence hereafter soybean system) are considered. The experiment was established in four replicates at all locations and years in a split plot design at Ames, Morris, and Rosemount but a randomized complete block design (RBCD) was used at Prosper. Details of plot sizes are reported in a previous publication [33].

Treatments include three cover-crop interseeding dates (R4, R5, and R6 growth stages for maize, and R6, R7, and R8 growth stages for soybean). The three cover crops were winter camelina (cv. Joelle), pennycress (line, MN106), and winter rye (cv. Rymin). A control with no-cover crop (business as usual) was also included as a treatment. Interseeding date was the main plot, cover crops and no-cover crop (control) assigned as the subplot for the split plot design. The treatments for the RCBD were factorial combination of interseeding date by cover crops plus a control (no-cover crop). Our previous report on the establishment of these cover crops [33] showed that interseeding dates did not have significant effects on cover-crop biomass yield and we assume it is likely that they did not have an effect on soil N. Therefore, to see the effects of cover crops only, soil and cover-crop aboveground biomass samples for laboratory testing were collected from only the R5 interseeding date for maize and R7 for soybean to save time and minimize costs associated with sample collection, sample processing, and testing. These R5 and R7 cover-crop interseeding dates were between the first and second week of September depending on location and year [33]

### 2.3. Seeding, Plot Management, and Data Collection

Maize and soybean were planted with 76-cm row spacing at each location. Each plot contained four rows of maize or soybean. Prior to seeding both maize and soybean, spring tillage was performed (type of tillage and depth varied by location). Detailed management practices used are described in a previous publication [29]. In brief, WC, PC, and WR were broadcast interseeded at 1368, 1064, and 222 pure live seeds (PLS) m$^{-2}$. In the following spring, soybean was relay planted into PC and WC using recommended varieties and seeding rates in the different locations. The dates of relay soybean seeding varied by location but were between the first and second week of May. Winter rye was terminated with glyphosate (N-(phosphonomethyl) glycine) at 1.1 kg a.i. ha$^{-1}$ about one week before seeding soybean in the spring, but PC and WC were grown to maturity and harvested for seed between late June and early July (depending on location and year).

In autumn (around the first week of November) and in the spring (from the last week of April to first week of May), cover-crop biomass samples were collected by clipping the plants at the soil surface from two separate 0.09 m$^2$ areas from the first and third rows of each plot for all locations except in Ames where two 0.76 m$^2$ areas were used. Cover-crop biomass and soil sampling varied by year and location, but were around the first week of November, before soil freeze, for autumn data and between the last week of April and first week of May for spring data. The cover-crop biomass samples were oven-dried at 65 °C (until constant mass was achieved) and ground to pass through 0.45 mm sieves. Then, 0.2 g of this plant sample was used to determine C and N using a dry combustion method (LECO CN828, LECO Corporation, St. Joseph, MI, USA). The concentration of C and N in cover-crop biomass was converted to kg ha$^{-1}$ using the respective cover crop's dry biomass yield before statistical analysis. Carbon and N in cover-crop biomass samples collected in autumn 2017 were not measured due to contamination of biomass samples with soil.

Soil samples from 0–15 and 15–60 cm soil depths were collected using a soil probe with 1.7-cm inside diameter (JMC Soil Samplers, Newton, IA, USA) from the center row of each plot and two composites per plot were taken. Soil samples were taken at three different seasons (autumn, spring, and relayed soybean harvesting time). The autumn (AUT) and spring (SPR) soil samples were collected at the same time during cover-crop biomass sampling. In addition, soil samples were collected immediately following relayed soybean harvest (RSH) between the last week of September and the second week of October depending on year and location [33]. Soil samples were air dried until constant mass was achieved and ground to pass a 2 mm sieve. Then, 2 g of this soil was mixed with 20 mL of 1 M KCl, shaken for 1 h, and extracted through a Whatman number 42 filter paper. The extract was analyzed for $NH_4^+$-N and $NO_3^-$-N using automated Cd reduction and salicylate methods [34] on a continuous flow-injection analyzer (Lachat Quik Chem 8500, Hach Company, Loveland, CO, USA).

*2.4. Statistical Analysis*

Analysis of variance was performed using PROC GLIMMIX in SAS 9.4 [35]. Data from the two sets (maize system and soybean system) were analyzed separately. In addition, data were combined over years but analyzed separately for each location and season. Cover crops were fixed effect, but replication and interaction with cover crops were random effects. Treatment means were separated by LSD at $p = 0.05$ when ANOVA showed a significant difference of $p \leq 0.05$. PROC CORR in SAS was used for Pearson correlation analysis for selected parameters. Correlations were considered significant at $p \leq 0.05$. Cover-crop biomass C/N was calculated by dividing biomass C (kg C ha$^{-1}$) by biomass N (kg N ha$^{-1}$). At Rosemount, maize residues were not sufficiently removed and suppressed cover-crop growth, thus data for the maize system from this location were not included in the analysis.

## 3. Results

*3.1. Soil $NH_4^+$-N and $NO_3^-$-N Content*

Cover crops had no significant effect on soil $NH_4^+$-N content at 0–15 cm and 15–60 cm soil depths in either the maize or soybean systems at all locations (Table 2 and Figure 1a–d). At RSH, cover crops had no effect on soil $NO_3^-$-N at all locations for both soil depths (Table 2). Similarly, under the maize system, cover crops had no effect on soil $NO_3^-$-N at Ames in AUT and SPR (Table 2, Figures 2a and 3a). However, at Morris when data were compared over years, WC as well as WR decreased soil $NO_3^-$-N in the 15–60 cm soil depth by 36% in AUT compared with no-cover crop or PC (Figure 2b). In SPR at Prosper, PC and WR decreased soil $NO_3^-$-N significantly both at 0–15 and 15 to 60 cm soil depths when compared with no-cover-crop treatment (Figure 2c). In the soybean system, cover crops again had not effect on soil $NO_3^-$-N at Ames in both soil depths during all three soil sampling times (Figure 3a). However, at Morris, all cover crops in general decreased soil $NO_3^-$-N to the extent of 50% compared with no-cover-crop treatment in AUT at both the 0–15 and 15–60 cm soil depths (Table 2; Figure 3b). This decrease was also significant in the SPR but only for the 15–60 cm soil depth. At Prosper, the cover crops also resulted in less soil $NO_3^-$-N in AUT and SPR compared with the no-cover-crop treatment (Figure 3c). The decrease of soil $NO_3^-$-N in SPR in the 0–15 cm soil depth at Rosemount due to WR was 76% compared with the control treatment (no-cover crop) (Figure 3d).

**Table 2.** Analysis of variance table showing level of significance (*p*-values) for $NH_4^+$-N and $NO_3^-$-N (mg kg$^{-1}$) at 0–15 and 15–60 cm soil depths in autumn (AUT), spring (SPR), and relayed soybean harvesting time (RSH) at Ames, Morris, Prosper, and Rosemount when cover crops were interseeded in standing maize and soybean.

| Location | Maize System | | | | | | Soybean System | | | | | |
|---|---|---|---|---|---|---|---|---|---|---|---|---|
| | 0–15 cm | | | 15–60 cm | | | 0–15 cm | | | 15–60 cm | | |
| | AUT | SPR | RSH | AUT | SPR | RSH | AUT | SPR | RSH | AUT | SPR | RSH |
| | $NH_4^+$-N | | | | | | | | | | | |
| Ames, IA | 0.771 | 0.657 | 0.822 | 0.359 | 0.743 | 0.914 | 0.295 | 0.230 | 0.705 | 0.399 | 0.056 | 0.607 |
| Morris, MN | 0.686 | 0.707 | 0.854 | 0.929 | 0.259 | 0.829 | 0.937 | 0.055 | 0.768 | 0.819 | 0.650 | 0.788 |
| Prosper, ND | 0.100 | 0.993 | 0.089 | 0.129 | 0.168 | 0.349 | 0.543 | 0.373 | 0.695 | 0.318 | 0.603 | 0.736 |
| Rosemount, MN | NA | NA | NA | NA | NA | NA | 0.8703 | 0.637 | 0.148 | 0.085 | 0.393 | 0.067 |
| | $NO_3^-$-N | | | | | | | | | | | |
| Ames, IA | 0.838 | 0.481 | 0.669 | 0.209 | 0.162 | 0.593 | 0.288 | 0.143 | 0.892 | 0.752 | 0.165 | 0.891 |
| Morris, MN | 0.060 | 0.221 | 0.083 | 0.047 | 0.066 | 0.791 | 0.002 | 0.101 | 0.157 | 0.004 | 0.002 | 0.409 |
| Prosper, ND | 0.676 | 0.002 | 0.704 | 0.777 | 0.035 | 0.134 | 0.008 | 0.001 | 0.338 | 0.012 | 0.001 | 0.228 |
| Rosemount, MN | NA | NA | NA | NA | NA | NA | 0.127 | 0.025 | 0.492 | 0.037 | 0.0592 | 0.926 |

AUT = soil samples taken in autumn; SPR = soil samples taken in spring; RSH = soil sample taken immediately following relayed soybean harvesting; NA= Data not available.

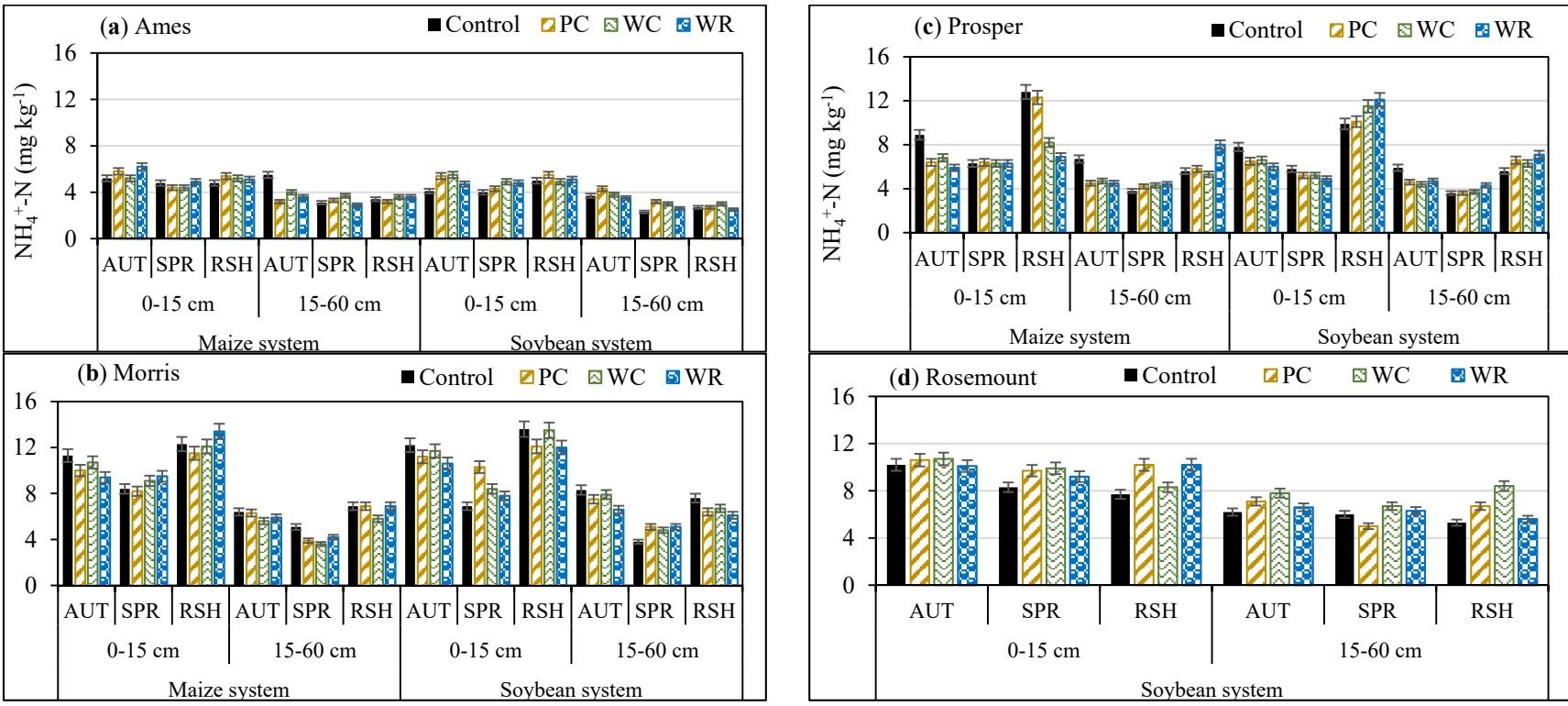

**Figure 1.** Mean soil NH$_4^+$-N (mg kg$^{-1}$) at 0–15 cm and 15–60 cm soil depths for control (no-cover crop), pennycress (PC), winter camelina (WC), and winter rye (WR) when cover crops were interseeded in standing maize and soybean at (**a**) Ames, (**b**) Morris, (**c**) Prosper, and (**d**) Rosemount. Data for Rosemount under maize are not included due to technical error. AUT = soil samples taken in autumn; SPR = soil samples taken in spring; RSH = soil sample taken immediately following relayed soybean harvesting. There was no statistical difference among treatments for same soil depth and same season in a system at a specific location. Error bars are standard error of the means.

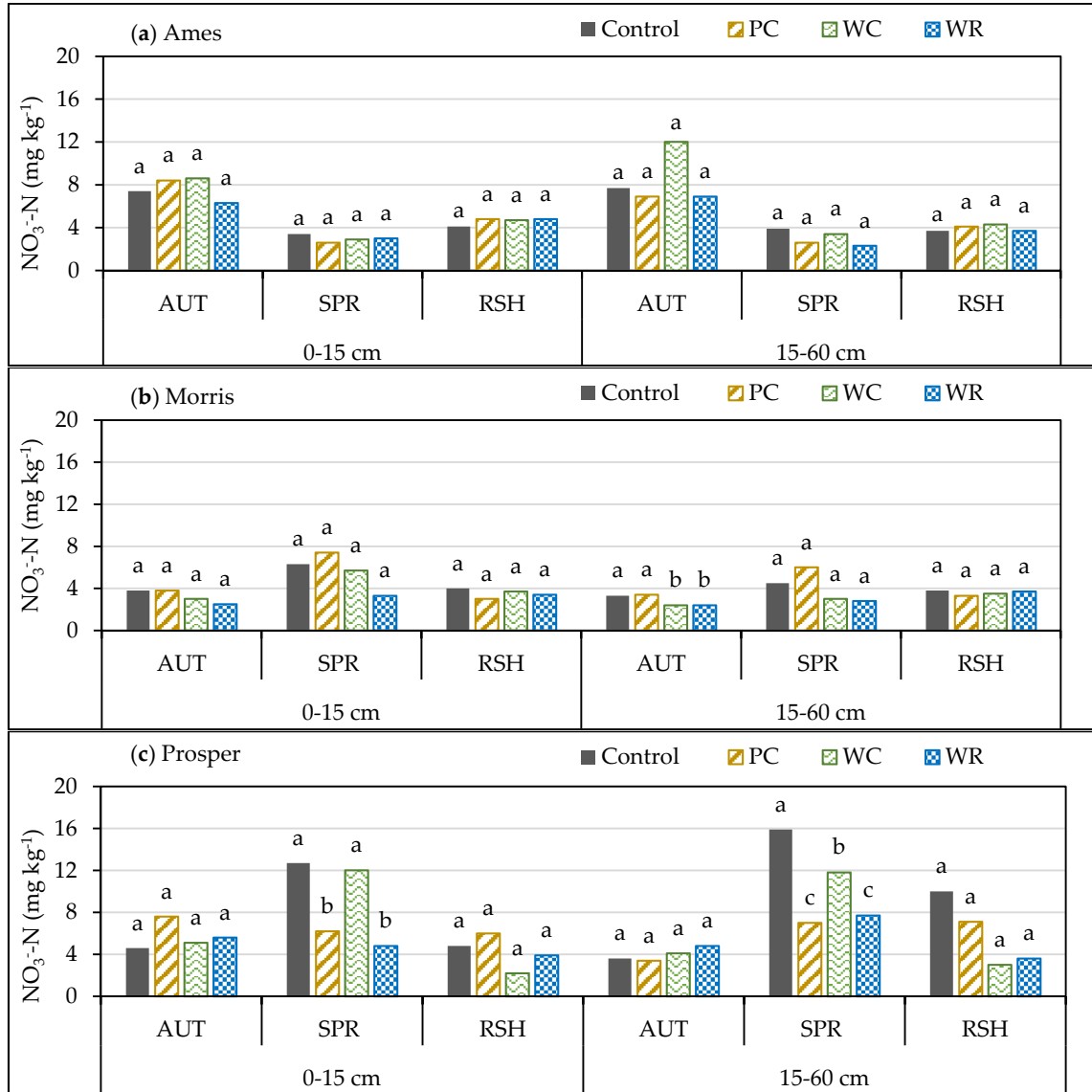

**Figure 2.** Mean soil $NO_3^--N$ (mg kg$^{-1}$) at 0–15 cm and 15–60 cm soil depths for control (no-cover crop), pennycress (PC), winter camelina (WC), and winter rye (WR) when cover crops were interseeded in standing maize at (**a**) Ames, (**b**) Morris, and (**c**) Prosper. AUT = soil samples taken in autumn; SPR = soil samples taken in spring; RSH = soil sample taken immediately following relayed soybean harvesting. Means with the same letter under same soil depth and same season at a location were statistically the same.

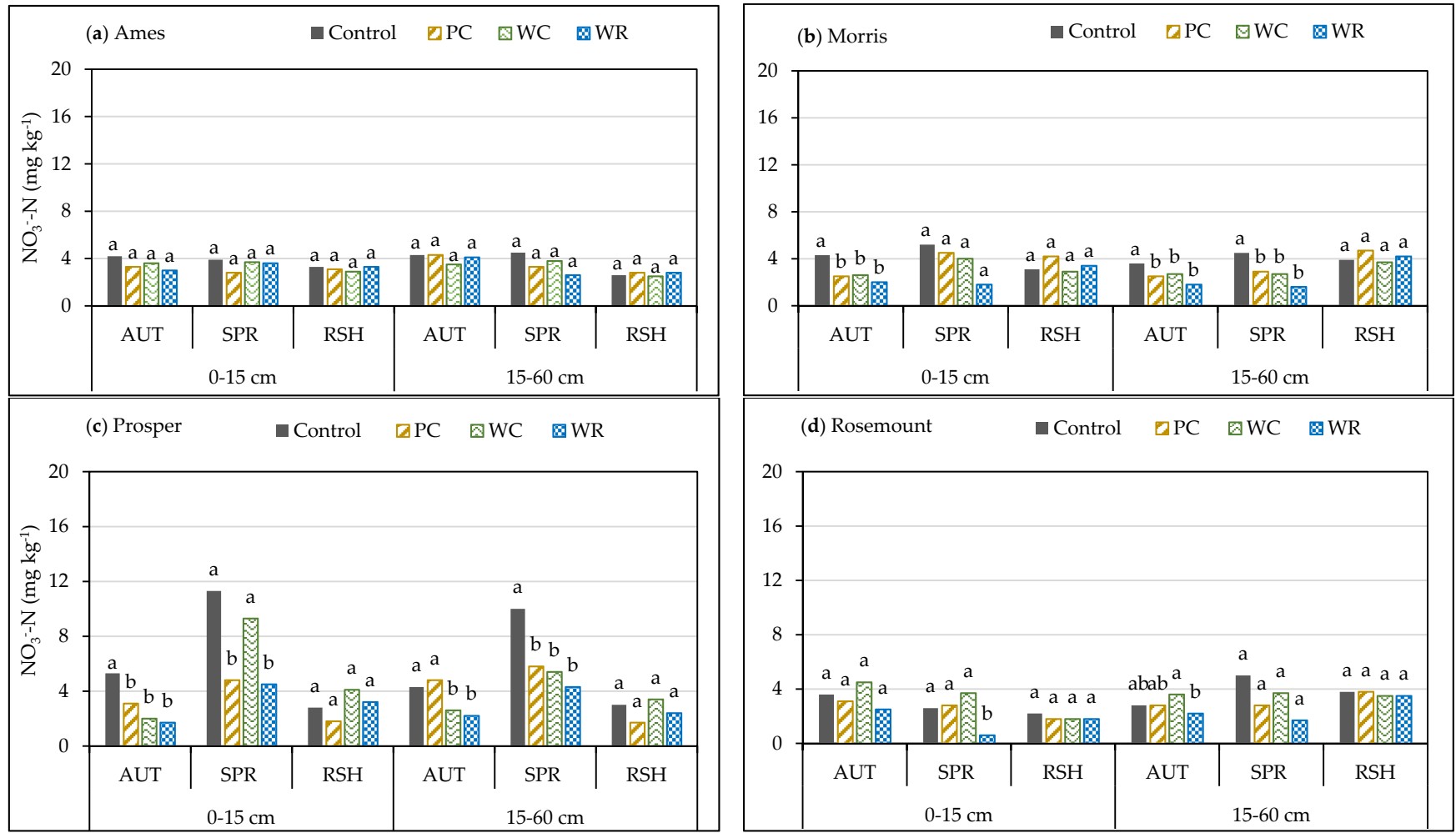

**Figure 3.** Mean soil $NO_3^--N$ (mg kg$^{-1}$) at 0–15 cm and 15–60 cm soil depths for control (no-cover crop), pennycress (PC), winter camelina (WC), and winter rye (WR) when cover crops were interseeded in standing soybean at (**a**) Ames, (**b**) Morris, (**c**) Prosper, and (**d**) Rosemount. AUT = soil samples taken in autumn; SPR = soil samples taken in spring; RSH = soil sample taken immediately following relayed soybean harvesting. Means with the same letter under same soil depth and same season at a location were statistically the same.

### 3.2. Nitrogen and Carbon Accumulation in Cover-Crop Biomass

For both cropping systems and all locations, WR nearly always accumulated more biomass N than either WC or PC (Table 3 and Figure 4a,b) during autumn, except for the soybean system at Prosper. However, by spring, the amount of N accumulated in WC or PC biomass was for the most part the same as WR in both cropping systems, with the exceptions of the maize system at Ames (Figure 4a) and the soybean system at Rosemount (Figure 4b).

**Table 3.** Analysis of variance table showing level of significance (*p*-values) of cover crop biomass nitrogen and carbon accumulation in autumn and spring at Ames, Morris, Prosper, and Rosemount when cover crops were interseeded in standing maize and soybean.

| Variables | Ames | | Morris | | Prosper | | Rosemount | |
|---|---|---|---|---|---|---|---|---|
| | Autumn | Spring | Autumn | Spring | Autumn | Spring | Autumn | Spring |
| **Maize System** | | | | | | | | |
| Biomass nitrogen | 0.002 | 0.010 | 0.001 | 0.370 | 0.008 | 0.332 | NA | NA |
| Biomass carbon | 0.001 | 0.008 | 0.001 | 0.142 | 0.005 | 0.104 | NA | NA |
| **Soybean System** | | | | | | | | |
| Biomass nitrogen | 0.027 | 0.106 | 0.004 | 0.295 | 0.216 | 0.238 | NA | 0.003 |
| Biomass carbon | 0.027 | 0.032 | 0.000 | 0.118 | 0.062 | 0.069 | NA | 0.004 |

NA = data not available.

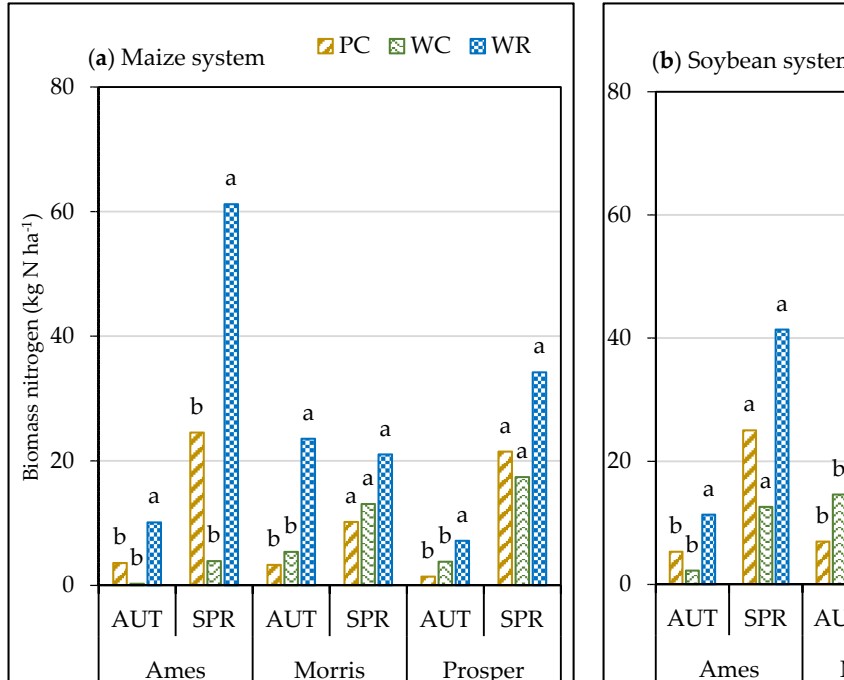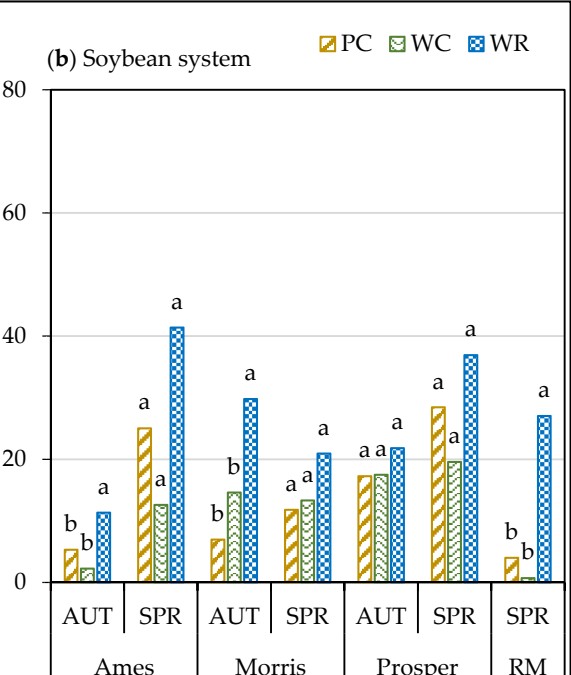

**Figure 4.** Mean nitrogen accumulation (kg N ha$^{-1}$) for the different cover-crops (PC = pennycress, WC = winter camelina, and WR = winter rye) biomass in autumn (AUT) and spring (SPR) at Ames, Morris, Prosper, and RM (Rosemount) when cover crops were interseeded in standing (**a**) maize and (**b**) soybean. Means with same letter within a location and a season for the same system were statistically the same. Data for maize system and in autumn for soybean system for Rosemount (RM) are not available.

Similarly, WR accumulated more biomass C than PC or WC in autumn both in the maize and soybean systems (Table 3; Figure 5a,b). This difference persisted during the spring at Ames for both cropping systems and to some extent at Prosper under the maize system. However, at the two most northerly sites, Morris and Prosper, there was no difference in C accumulation among the cover crops in the spring, except for the maize system at Proposer where WR accumulated more C than WC (Figure 5a).

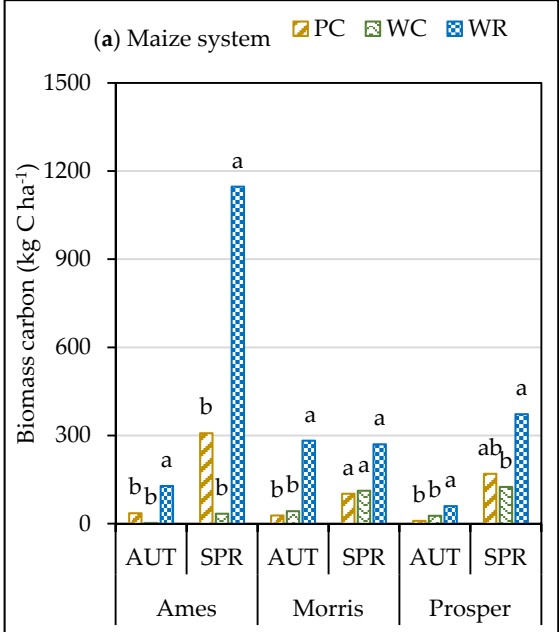 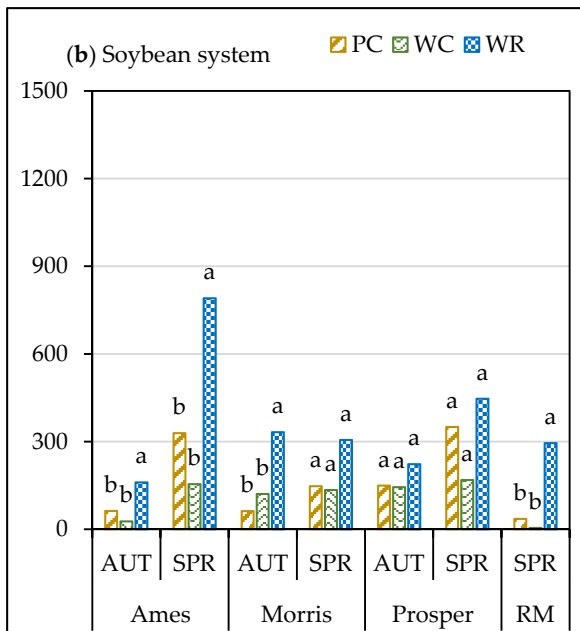

**Figure 5.** Mean carbon accumulation (kg C ha$^{-1}$) in the different cover-crop (PC = pennycress, WC = winter camelina, and WR = winter rye) biomass in autumn (AUT) and spring (SPR) at Ames, Morris, Prosper, and RM (Rosemount) when cover crops were interseeded in standing (**a**) maize and (**b**) soybean. Means with same letter within a location and a season for the same system were statistically same. Data for maize system and in autumn for soybean system at Rosemount (RM) are not available.

### 3.3. Carbon to Nitrogen Ratio (C/N) in Cover-Crop Biomass

The difference in cover-crop biomass C/N (based on content) was significant under maize and soybean systems both in autumn and spring (data not shown) with greater C/N for WR followed by PC compared with WC. As it can be calculated from Figures 4 and 5, WR had greater C/N than PC or WC at all locations both in maize as well as soybean systems in both autumn and spring. For instance, the mean C/N for the different cover crops when averaged over locations under the maize system in spring were 10, 8, and 13 for PC, WC, and WR, respectively. The corresponding C/N for soybean system were 11, 9, and 14 for PC, WC, and WR, respectively.

## 4. Discussion

### 4.1. Soil $NH_4^+$-N and $NO_3^-$-N

One of the major reasons for growing a cover crop is to scavenge leftover soil mineral N following conventional crop harvest (e.g., maize and soybean), thus reducing the potential for unwanted N movement into ground or surface waters. In the present study, a decrease in soil-available N ($NH_4^+$-N and $NO_3^-$-N) in the soil profile was expected due to cover-crop N uptake compared with a no-cover (business as usual) treatment. However, this was not observed for $NH_4^+$-N at all locations and seasons under both maize and soybean systems. This is likely due to the cover crops' preferential uptake of $NO_3^-$-N over $NH_4^+$-N. Many growing plants prefer $NO_3^-$-N over $NH_4^+$-N [36]. Furthermore, other research results show that plant preference for $NO_3^-$-N depends on its physicochemical nature [37].

Soil $NO_3^-$-N is an anion and usually soluble and available in soil water, which favors increased plant uptake, thus decreasing its concentration in the soil. Conversely, $NH_4^+$ is a cation which can bind to clay minerals, potentially making it less available for plant uptake. This may have played a factor in the present study and is a likely reason why cover crops had little or no effect on soil $NH_4^+$-N levels.

Soil $NO_3^-$-N depletion was demonstrated at three out of four locations in response to the cover crops grown in this study. The absence of significant differences in soil $NO_3^-$-N at Ames, regardless of cropping system, remains an enigma. Because of the large amount of N accumulation found in this study, coupled with high biomass production [33], WR at Ames might be expected to reduce soil $NO_3^-$-N, but this was not found to be the case. Previous research results showed that cover crops with >390 kg ha$^{-1}$ spring biomass reduce soil $NO_3^-$-N compared with fallow [18]. However, this conclusion does not agree with our findings for Ames where more than 390 kg ha$^{-1}$ WR biomass was produced in spring [33]. A potential reason for the lack of significant soil $NO_3^-$-N changes at Ames in our study despite cover-crop growth, could be related to N mineralization. With the relatively wet and warm weather at Ames (Table 1), conditions may have been prime for soil N mineralization from soil organic matter, which also may have been differentially affected by living cover. A previous study indicated improved net N mineralization with optimum temperature and soil moisture [38]. The higher temperature and relatively moist soil at Ames, could have resulted in improved net mineralization, thus avoiding soil $NO_3^-$-N depletion due to crop uptake compared with the fallow. Another possible reason could be the already-low (insignificant) amount of soil $NO_3^-$-N as shown in the results which was generally less than 4 mg kg$^{-1}$ particularly in spring. We also think that if soil sampling was done more than one time (which is the limitation of this study) to see soil $NO_3^-$-N dynamics, a different result could have been obtained but further research is needed.

The decrease of soil $NO_3^-$-N due to WC and PC at Morris and Prosper may indicate that adoption of these winter oilseed covers at more northerly latitudes across the Corn Belt could enhance mitigation of water contamination by $NO_3^-$-N while increasing crop diversity of the maize–soybean rotation. Previously, it was demonstrated that when WC and PC were established as cover crops by direct-drilling after spring wheat (*Triticum aestivum* L.), these over-wintering oilseeds reduced soil $NO_3^-$-N by as much as 68% compared with winter fallow controls, and this was similar to results for winter rye [29]. The amount of N accumulated by the winter oilseeds ranged from 28 to 49 kg N ha$^{-1}$ [29], which is considerably more than the 1.9 to 19.4 and 2.4 to 24.0 kg N ha$^{-1}$, under the maize and soybean systems, respectively, measured in our study. The challenges associated with WC and PC growth following interseeding into standing crop [33] may limit their impact on sequestering excess soil N as compared with cropping systems where direct-seeding methods are used for establishment. Research is warranted to study the yield and ecosystem services tradeoffs of employing early harvest of maize and soybean to promote direct-drilling of winter oilseeds in autumn.

This study demonstrated that interseeded PC and WC generally did not perform as well as WR in reducing soil $NO_3^-$-N or sequestering N in its biomass. This difference between WR and the oilseeds is not a surprise because these oilseeds are relatively-new crops and have not received nearly as much research and development as WR into developing better and improved cultivars. Therefore, additional research is warranted to improve PC and WC genetics for enhanced establishment when interseeded into standing maize and soybean to provide soil cover and diversify the maize–soybean-dominated cropping systems.

### 4.2. Nitrogen and Carbon Accumulation in the Biomass

The greater C and N accumulation in WR biomass was partly due to its greater biomass yields compared with PC or WC [33]. A recent study on cover crops indicated that tissue N accumulation in PC was less than WR [18]. Another study showed that WR had greater N accumulation in the biomass than WC [39] and our results support these findings. In our study, WC at Morris showed relatively-low N accumulation in spring compared with a previous report when WC was interseeded around "silking stage" of maize for the same location (10 vs. 59 kg N ha$^{-1}$) [30]. This could be due to differences in

biomass yield caused by weather variability within a location. As shown in Table 1, year-to-year variation in precipitation and temperature is clear. For instance for Ames, the first growing season (2016/2017) was relatively warmer and wetter compared with the long-term average. Our results showed that N accumulated in cover-crop biomass was negatively and significantly correlated (data not shown) with soil $NO_3^-$-N. This indicates the contribution of cover crops to enhance soil $NO_3^-$-N uptake to incorporate soil N into their biomass and thus potentially minimizing nitrate leaching and runoff and promote nutrient cycling.

*4.3. Carbon to Nitrogen Ratio (C/N) in Cover-Crop Biomass*

Cover crop C/N influences microbial activities and N microbial immobilization during biomass decomposition. A balanced proportion of C and N, in addition to the quality of C in cover-crop biomass, is important for healthy soil biological activity when cover crops are returned to the soil. This is particularly essential in relation to the release of nutrients such as N from cover-crop biomass decomposition and mineralization. A previous study showed that the best predictors of N mineralization from leaves and stems of cover crops is C/N [40]. Crop residue decomposition and N mineralization is considered rapid when C/N is less than 20:1 [41]. The cover-crop biomass in the two cropping systems in our study had C/N with mean values less than 20:1, indicating the possibility for optimum mineralization if these cover crops are incorporated into the soil. A previous study indicated that cover-crop C/N is another determinant of ecosystem services that is positively related to nitrate-leaching prevention [42] and our results (a negative but significant correlation of C/N with soil $NO_3^-$-N) (data not shown) support this finding. Overall, in our study, the C/N results in the maize system were lower than in the soybean system. Soybean, being a nitrogen fixer due to its association with soil bacteria, was supposed to have less C/N than the maize system. However this was not the case. This could have been due to a greater N uptake of cover crops from the soil in the maize system (due to a greater residual N resulting from prior maize N fertilization) than in the soybean system. For instance for Ames, the control from the maize system in autumn had more soil $NH_4^+$-N + $NO_3^-$-N (12.6 mg $kg^{-1}$) in the 0–15 cm soil depth than the soybean system (8.3 mg $kg^{-1}$).

## 5. Conclusions

The contribution of interseeded cover crops in reducing soil $NO_3^-$-N compared with no-cover crop (winter fallow treatment which is business as usual) was substantial, particularly for WR, which decreased soil $NO_3^-$-N to the extent of 76% compared to fallow. Winter camelina at Morris and PC at Prosper also reduced soil $NO_3^-$-N. This indicates that PC and WC have potential in reducing water pollution at more northerly latitudes, at least when employed in the interseeding systems used in this study. Generally, across locations and years, PC and WC accumulated less C and N in their biomass compared with WR. Developing PC and WC cultivars with improved performance under interseeding conditions could help enhance crop diversification and N use in the maize–soybean rotation that is most prevalent in the upper Midwest US. In the near-term, developing agronomic practices other than interseeding to integrate these winter oilseeds into maize and soybean systems may be a key strategy to improve nutrient cycling, water quality, and enhance crop diversification.

**Author Contributions:** Conceptualization and methodology, R.W.G., A.W.L., B.L.J., M.S.W., F.F., and M.T.B.; data compilation, analysis, and original draft preparation, Y.A.M.; review and editing, Y.A.M., R.W.G., A.W.L., B.L.J., M.S.W., F.F., M.T.B, S.P., and H.L.M.; project supervision, R.W.G. and M.T.B. All authors have read and agreed to the published version of the manuscript.

**Funding:** This research was funded by USDA-National Institute of Food and Agriculture-Coordinated Agricultural Program (Grant number: 2016-69004-24784).

**Acknowledgments:** The authors are grateful to Jane Johnson for reviewing the original draft and Alex Hard, Chuck Hennen, Dahlia Whiting, Dean Peterson, Grace Laskey, Janeille Schaubhut, Jay Hanson, Jim Eklund, Joe Boots, Matt Thom, Paula Peterson, Roger Hintz, Scott Larson, and Taylir Bullick for their help in field-plot management, data and sample collection, sample processing, and testing. We thank the three anonymous reviewers for their input and comments.

**Conflicts of Interest:** The authors declare that there is no conflict of interest.

## Abbreviations

| | |
|---|---|
| C | carbon |
| AUT | autumn |
| N | nitrogen |
| PC | pennycress |
| RSH | relayed soybean harvesting time |
| SPR | spring |
| WC | winter camelina |
| WR | winter rye |

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
