# Peer review of "Soil Nitrogen in Response to Interseeded Cover Crops in Maize–Soybean Production Systems"

_agronomy, doi:10.3390/agronomy10091439_

Round 1
Reviewer 1 Report
see the attached file

Author Response
Reviewer #1:
Thank you very much for reviewing our manuscript. Please see point by point replies (attached) for your comments and inputs. Correction made following your review are highlighted yellow as shown in the updated version of the manuscript. Line numbers are shown in this attachment where correction are made to make it easy for cross check.
Sincerely,

Reviewer 2 Report
please see Attachment

Author Response
Reviewer #2:
Thank you very much for reviewing our manuscript. Please see point by point replies (attached) for your comments and inputs. Correction made following your review are highlighted yellow as shown in the updated version of the manuscript. Line numbers are shown in this attachment where correction are made to make it easy for cross check.
Sincerely,

Reviewer 3 Report
Overall the paper is generally well written but seems incomplete. Discussion of the results and comparison to other published works is limited . The experimental design of the study is sound, but the presentation of the results is weak. Look at other published articles from the journal to review how figures should be prepared. Please note that line numbers reset at Figure 2 so reviewer suggestions will follow line numbers as given. It is up to the authors to find associated comments and make corrections.
Lines 18-19: Scientific names for crop species are not consistent. Provide consistent strategies for listing scientific names.
Line 23: Change “to the extent” to “up to”
Line 30: These keywords are not reflective of the objective of the paper. Consider revising.
Line 42: This standard is for water, identify it as such.
Line 43: Why have the nitrate loads increased in that given time?
Lines 56-57: Cost share by whom?
Lines 64-66: Provide citation.
Lines 84-85: List all in-text citations according to the Journal instructions.
Table 1. Is the “2016/2018” weather data an average for those three years? If so, the data needs to be presented on a monthly basis for each year and not averaged.
Lines 110-111: How does collecting samples from those dates achieve your objective? It does not appear to support that idea as if that is your primary objective then all planting dates should be included.
Lines 124-126: More information on biomass collection is needed. How much area was collected from? How many reps per plot? Unable to assess the quality of your methods with this amount of information.
Lines 124-131: List the actual sampling dates for each cropping season.
Lines 132-134: Was this enough soil samples to represent the plot? You also need to describe the site soil with far more detail. That will have a significant impact on your results and discussion.
Lines 156-160: This entire paragraph should be removed. At best it might be included in the discussion, but certainly not the results section.
Section 3.1 Soil Ammonium and Nitrate Content: Significantly more discussion is necessary. It should center around concerns regarding water quality from runoff and leaching which is not discussed at all in your results.
Lines 166-167: How did you determine the decrease in nitrate by 36%? Did you compare across years?
Figure 1: What do the error bars represent? Your x-axis is confusing. Which cropping system do these describe?
Line 182: You say there is no data for the Rosemount maize system included but there is a set of data included in the figure which I assume is for the soybean system. This needs to be clarified as it is very confusion right now.
Table 3: Unbold “Biomass nitroge.”
Line 209: Incomplete statement
Table 3: Where are the biomass production values? They are not included anywhere in the manuscript but play an important role in the results and discussion.
Figure 2: Figure needs error bars. Be consistent with lettering to signify differences. Figure description needs editing for fonts and removal of the repeated second sentence.
Figure 3: Similar errors as figure 2 that need to be addressed.
Line 2 after figure 3: Insert “biomass” before N
Figure 4: Needs error bars.
Figure 5: Needs error bars.
Line 26: Does not belong in results section. Move to discussion.
Line 31: Clarification required. Are you saying that cover crops were not significantly different based on locations?
Line 36: Replace “pick up” with “scavenge”
Line 42: Does these chosen cover crops show evidence of preference for one N fraction over another?
Line 52: Not the correct style for the journal. Fix all instances.
Line 53-55: There needs to be more citations showing a similar trend. Reference those materials in your discussion to strengthen your argument.
Line 56: Where is this data? Needs to be included to strengthen the discussion.
Lines 60-61: Wouldn’t we see that result in your ammonium values? We didn’t, so what else could be happening to the N aside from mineralization?
Line 70: Incorrect in-text citation.
Line 71: Add “N” in the kg ha-1
Line 79-80: Why did a lack of past research have an impact on the current research project?
Lines 90-91: Why speculated about the weather’s impact? You have the data, include an analysis of this.
Line 92: Why is that data not included? It could strengthen the arguments of your manuscript.
Lines 96-108: What about the structural components of the biomass? They often have a more significant impact on decomposition compared to C:N. What evidence exists in the literature to support that? Especially since you are comparing to different types of cover crops.
Lines 107-108: Do you have evidence to support this argument? If so, it definitely needs to be included.
References: Wildly inconsistent and needing much work.
Author Response
Reviewer #3:
Thank you very much for reviewing our manuscript. Please see point by point replies (attached) for your comments and inputs. Correction made following your review are highlighted yellow as shown in the updated version of the manuscript. Line numbers are shown here correction are made to make it easy for cross check.
Sincerely,

Round 2
Reviewer 3 Report
The authors did an excellent job addressing my comments. I commend them for their timely response . Good luck with your future endeavors!